# Preliminary Toxicity Evaluation of a Porphyrin Photosensitizer in an Alternative Preclinical Model

**DOI:** 10.3390/ijms24043131

**Published:** 2023-02-05

**Authors:** Miryam Chiara Malacarne, Maristella Mastore, Marzia Bruna Gariboldi, Maurizio Francesco Brivio, Enrico Caruso

**Affiliations:** 1Department of Biotechnology and Life Sciences (DBSV), University of Insubria, Via J.H. Dunant 3, 21100 Varese, Italy; 2Department of Theoretical and Applied Sciences (DiSTA), University of Insubria, 21100 Varese, Italy

**Keywords:** photodynamic therapy, 5,10,15,20-tetrakis(4-hydroxyphenyl)-porphyrin, *G. mellonella*, phototoxicity, cytotoxicity, cellular uptake

## Abstract

In photodynamic therapy (PDT), a photosensitizer (PS) excited with a specific wavelength, and in the presence of oxygen, gives rise to photochemical reactions that lead to cell damage. Over the past few years, larval stages of the *G. mellonella* moth have proven to be an excellent alternative animal model for in vivo toxicity testing of novel compounds and virulence testing. In this article, we report a series of preliminary studies on *G. mellonella* larvae to evaluate the photoinduced stress response by a porphyrin (PS) (TPPOH). The tests performed evaluated PS toxicity on larvae and cytotoxicity on hemocytes, both in dark conditions and following PDT. Cellular uptake was also evaluated by fluorescence and flow cytometry. The results obtained demonstrate how the administration of PS and subsequent irradiation of larvae affects not only larvae survival rate, but also immune system cells. It was also possible to verify PS’s uptake and uptake kinetics in hemocytes, observing a maximum peak at 8 h. Given the results obtained in these preliminary tests, *G. mellonella* appears to be a promising model for preclinical PS tests.

## 1. Introduction

Photodynamic therapy (PDT) is a clinically approved, highly selective and minimally invasive medical procedure for the treatment of various diseases including cancers [1]. To be applied, PDT requires the topical or systemic administration of a photosensitive molecule generally called photosensitizer (PS), the irradiation of the targeted area with an appropriate light wavelength [2] and molecular oxygen (O_2_) [3]. Although individually non-toxic, together they give rise to a photochemical reaction that generates reactive oxygen species (ROS) [4].

The absorption of light radiation activates the electrons of the PS, which are then transferred from a fundamental electronic state to a more energetic state called the excited electronic state (1PS*). From this, through intersystem crossing (ISC) and transfer of energy to molecular oxygen (O_2_) [5] or to substrates [6] such as the cell membrane, species are formed, such as singlet oxygen, which lead to cell death, damage to the tumor vascular system and, in some cases, a local inflammatory reaction that can give rise to systemic immunity [7,8].

Among the different classes of PSs [9,10] studied and analyzed for PDT application, the largest class is represented by porphyrin [11].

Over the years, the use of animal models in biomedical research has made it possible to understand human pathophysiology and, significantly, implement vaccines and antibiotics development [12].

As scientific progress has advanced, the use of animals in research has increased. For this reason, it is extremely important to keep in mind the “three Rs of animal research” [13,14,15]. With this simple concept, the scientific community underlines the need to reduce, refine and, if possible, replace animals in research. Nowadays, in biomedical research, approximately 20 million animals are used, dominated by mouse and rat models due to their physiological and immune systems that are nearly equivalent to those of humans [16,17].

However, such animal models have significant logistical and ethical limitations [13] and the possible use of alternative models, especially invertebrates, offers significant advantages over mouse models [18,19].

In addition to the advantages of lower costs and ease of handling, invertebrates have an innate immune system extremely similar to that of mammals [17,20]. On the other hand, as regards the physiological aspect, insects cannot in any way be compared to mammals. Overall, it has been widely demonstrated that invertebrates, particularly nematodes and insects, can be a valid alternative model for preclinical studies on pharmacological molecules and for toxicity studies [21,22,23].

Among various insects, *Galleria mellonella*, a lepidopteran also known as the Greater Wax Moth (GWM) [24,25], which in nature is a parasite of beehives [17,26], has, since the beginning of the 21st century, been used for medical and scientific research [17,27,28,29,30,31].

In an increasing number of studies, *G. mellonella* has been validated as an alternative invertebrate model to evaluate the pathogenetic mechanisms of various human pathogens and their interaction with the immune system [27,28,29], or to evaluate new therapeutic treatments for infectious diseases [17,30,31]. *G. mellonella* has a typical holometabolous cycle, with life stages occurring at a temperature close to that of mammals, since it survives without physiological changes at 37 °C [26,32]. Several physiological characteristics, such as morphology, size, easy handling and economy of breeding, make this invertebrate an extremely useful animal model in preclinical in vivo experimentation.

Insects possess an innate (lacking B-cell immunoglobulin) but highly effective immune response; their immunity is based on the synergistic action of molecular systems (humoral immunity) and immunocompetent cells (cellular immunity), able to react against compounds and non-self-biotic invaders [17,33,34].

Humoral immunity supports cells and consists of molecular systems, such as the prophenoloxidase–phenoloxidase (proPO) system, which is responsible for melanin formation in the humoral encapsulation process, antibacterial peptides (AMPs) and lysozyme, which are responsible for bacterial clearance [35,36]. Cellular decences are mainly carried out by hemocytes, present in the hemolymph (insect blood); they play a key role in defense processes such as phagocytosis, nodulation and encapsulation, which are key processes in the defense against external agents [33,34].

Like many blood cells, hemocytes are extremely reactive and sensitive to the presence of foreign molecules; therefore, assessing their viability and ability to absorb and interact with administered drugs allows them to be used as markers of physiological conditions [17,33,34,37].

In this work, we evaluated the photodynamic effect of a PS, belonging to the porphyrin family, on an invertebrate model which is validated for toxicological and imunological trials [38,39,40]. The chosen molecule was 5,10,15,20-tetrakis(4-hydroxyphenyl)-porphyrin (also called TPPOH) (Figure 1), a commercial porphyrin, whose photostability has already been studied in the past and whose photodynamic activity on tumor cells is known [41,42]. The survival over time of insect larvae was evaluated after the administration of decreasing concentrations of TPPOH, both in the dark and after light irradiation. The cell viability was determined at various times after administration of increasing TPPOH concentrations. Finally, the time course of TPPOH cellular uptake was also monitored by both flow cytometry analysis and fluorescence microscopy.

## 2. Results

### 2.1. TPPOH Toxicity in G. mellonella Larvae

First, DMSO and PBS potential toxicity were evaluated by injection into *G. mellonella* larvae. These control tests showed that both DMSO (at the highest concentration) and PBS, used as solvents of TPPOH, did not affect the viability of larvae in either dark tests or after irradiation.

To evaluate TPPOH toxicity, larvae were injected with decreasing concentrations (from 1000 to 0.1 µM) of PS in PBS-DMSO and then kept in dark or subjected to 2 h irradiation. Under dark conditions, survival was evaluated at 24, 48 and 72 h after treatments (Figure 2); instead, for the assays under light condition, after 24 h from injection with PS, larvae were irradiated and survival was detected after further 24, 48 and 72 h (Figure 3).

As reported in Figure 2, lepidoptera well tolerate the compound in dark conditions. In the first 24 h after the injection, the overall survival rate is always greater than 80%, while, at longer time periods, a slight decrease can be seen, which in any case remains higher than 60% at both 48 and 72 h.

As shown in Figure 3, the larvae subjected to PDT showed almost complete mortality after 24 h at concentrations above 5 μM; in contrast, at concentrations below 5 μM, the larvae always showed a survival rate higher than 70%.

Based on the survival rate after photodynamic treatment, LD_50_ values were determined at three times (Table 1).

As showed in Table 1, the LD_50_ values are almost overlapping for 24 and 48 h, while it decreases at 72 h after irradiation without, however, showing a significant difference.

From the morphological point of view, we monitored changes in larva body after PDT (Figure 4); treatments induced variations in larvae morphology with a decrease in turgidity and progressive darkening which, over long periods (24–48 h), showed a blackish color due to the activation of the proPO system induced by stress and from ROS present in the hemolymph.

### 2.2. Cytotoxicity on Immunocompetent Cells (Hemocytes)

After assessing the phototoxic effects of TPPOH on G. mellonella larvae, the toxicity of PS was also evaluated on hemocytes. The viability of cells was assessed and quantified by cell count using Trypan blue exclusion test.

Figure 5 shows the viability rate of hemocytes from larvae injected with TPPOH and then incubated in the dark condition. Larvae injected with increasing concentrations of PS show a negligible cytotoxic effect; a weak toxicity was observed only at the highest concentration (1000 μM), and the decrease in cell number is time dependent.

After checking that no changes in hemocytes survival were observed in larvae kept in the dark, the experiment was repeated; in this case, 24 h after treatment, larvae were irradiated for 2 h (Figure 6).

It is not possible to determine the hemocytes population for all concentrations that resulted in larval mortality (1000–50 µM). In all other cases, no differences were observed between treated and control larvae.

### 2.3. Cellular Uptake: Flow Cytometry and Microscopy

TPPOH cellular uptake at different post-injection times (2, 8 and 24 h) has been evaluated by flow cytometric analysis, exploiting the intrinsic fluorescence of the PS.

Figure 7 shows an increase in cellular uptake up to 8 h post-injection. After this time, a decrease of fluorescence was observed.

The qualitative analysis of the cellular uptake, carried out by fluorescence microscopy, confirmed data obtained by flow cytometric analysis (Figure 7B).

### 2.4. ROS Detection after PDT Treatment

The presence of ROS was assessed by a slight modification of the nitroblue tetrazolium (NBT) reduction method of Glupov et al. [43]. NBT reacts with ROS to form a colored formazan derivative that can be monitored in various cell populations by bright-field microscopy.

The observed NBT reaction reflects ROS-generating activity in cell cytoplasm and can determine the ROS cellular origin in hemocytes. Hemocytes, treated with TPPOH and NBT under dark conditions or after light irradiation (PDT), were observed under microscopy (Figure 8B,C).

The presence of cytoplasmic precipitates is particularly evident in cells observed after PDT (Figure 8C); however, a weak reaction is also present in non-irradiated cells (Figure 8B). In contrast, no reaction is observed in irradiated hemocytes treated with NBT, without TPPOH (Figure 8A). To avoid interference of the proPO system present in hemocytes which, if activated, could increase ROS production [44,45], we added phenylthiourea (PTU), a phenoloxidase inhibitor, during cell separation and throughout the course of the test.

## 3. Discussion

PDT refers to a highly selective and poorly invasive therapeutic treatment approved for the treatment of some cancers. To be applied, PDT requires three main components: a photosensitive molecule (PS), a light source and molecular oxygen (O_2_) [1]. None of these will be toxic if taken individually but together they will give rise to a photochemical reaction that culminates in the production of reactive oxygen species (ROS) [2] of which the most important is certainly singlet oxygen (^1^O_2_) [3]. The result is cell death, damage to the tumor vascular system and, in some cases, a local inflammatory reaction can give rise to systemic immunity [4].

For the improvement of PDT, a determining factor is the search for new PSs that have greater activity and selectivity; consequently, the development of a large number of PSs is necessary to identify new models for in vivo tests that reduce costs and do not require all the complicated bureaucratic approvals of animal tests. In this context, experiments on alternative invertebrate models such as *G. mellonella* are becoming interesting.

In recent years, this lepidopteran has been used as a model for in vivo studies of host–pathogen interactions [36,39,46] and, consequently, for preclinical studies on the development of new antibiotics [33,47,48]. The well-established use of the *G. mellonella* model in experimentation confirms the scientific world’s intent to replace and/or reduce the use of vertebrates in preclinical trials [49]; moreover, being an invertebrate, *G. mellonella* does not fall under animal welfare regulations and guidelines.

Larvae of *G. mellonella* have been used in recent decades to investigate virulence and antimicrobial efficacy [50] and to obtain pharmacokinetic data such as clearance and maximum administrable drug concentration. These data can be correlated with those obtained in humans; in fact, numerous studies confirm that microbial pathogenicity and virulence determinants are the same in humans, mice and the Greater Wax Moth [51,52]. Overall, such physiological and functional aspects make this animal model extremely promising for both immunological and pharmacological studies.

The purpose of the present work was to evaluate the feasibility of using the larval stage of the lepidopteran *G. mellonella* as an alternative model to test the toxicity of a PS. The chosen PS belongs to the porphyrin family, which includes the largest number of PSs used in the antitumor and antibacterial fields [53,54,55,56,57].

The study foresees an initial assessment of the tolerability of the larvae at decreasing doses of PS in the absence of irradiation, since the drug must show activity only after appropriate light-activation [58,59].

As could be expected, no dark toxicity of PS was observed even at the highest concentrations and even at the longest incubation times (72 h).

The introduction of the irradiation step has completed the photodynamic process and should cause all those oxidative processes which, in cascade, cause cell death. Indeed, irradiation drastically changed the larval survival outcomes. Up to the concentration of 10 µM, the larvae were all dead already 24 h after the end of the irradiation step. Instead, from the concentration of 5 µM, a predominant larval survival was observed and, in this case, a significant correlation between the administered concentration and treatment time was not observed. The latter result showed that the photodynamic effect is obtained in the first 24 h, without showing a further significant increase in mortality in the following 48 h.

From the data, it is possible to obtain the dose–response curves which allow us to obtain the LD_50_ values which are around 5 µM. The non-statistically significant difference confirms, as already observed in Figure 3, that the post-irradiation incubation time does not affect larval survival.

Another important aspect to consider about the physiology of *G. mellonella* is the functional homology of its immune system with that of vertebrates; although it lacks an adaptive immune response, the immune system of *G. mellonella* shares many similarities with the mammalian innate immune response. Defense processes such as the proPO system [60,61], the Toll and Imd activation pathways [62], lysozymes [63], cells with phagocytic/cytotoxic activity and granuloma-like cell encapsulation [64] are all aspects of the invader discrimination or elimination phases functionally comparable to vertebrates’ innate defenses. It is therefore important to also evaluate the effects of PS (dark and light toxicity) on the immunocompetent cells of *G. mellonella*; thus, hemolymph was collected, and the cellular component was isolated to determine its viability.

In the dark, a partial cytotoxic effect is observed only at the maximum concentration (1000 µM), with a lethality of about 20% at 24 and 48 h and 45% at 72 h. Administration of concentrations lower than 1000 µM had no effect on the viability of immunocompetent cells. After the irradiation step, hemocyte viability can never be calculated above 10 µM due to the complete mortality of larvae. In all cases where larval survival was observed, hemocyte viability was comparable to that of control (untreated larvae) and was always above 90%, suggesting that PDT does not affect the viability of immunocompetent cells. 

Since no significant change in hemocyte viability is observed, we tested whether PS penetrated immunocompetent cells or was distributed in the humoral component of hemolymph. To this end, larvae were treated with 100 µM of PS and, at fixed times (2, 8 and 24 h), uptake was evaluated through qualitative and quantitative analysis, by both fluorescence microscopy and flow cytometry. From the data obtained, it is observed that PS is localized exclusively in the cellular component of the hemolymph while none is observed in the acellular component. Furthermore, the absorption reached a maximum peak 8 h after the injection, while at 24 h, the uptake was reduced by about 50% compared to the previous one. The lower fluorescence detected at long times is probably linked to the PS’s degradation or elimination from hemocytes.

To assess the oxidative stress induced by TPPOH and PDT, we analyzed the presence of reactive oxygen species (ROS) in *G. mellonella* hemocytes. As reported in the literature on different cell populations [65], the formation of intracellular ROS can be assessed directly by microscopy; in particular, Glupov [43] showed the presence of ROS using NBT after bacterial molecule-induced stress in *G. mellonella* hemocytes.

Our results showed that treatment with TPPOH followed by irradiation induced ROS production by showing NBT-derived formazan salts within the cytoplasm due to a marked level of cellular stress. Hemocytes treated with TPPOH without PDT lead to a slight production of ROS.

The results obtained suggest that *G. mellonella* could be a possible alternative animal model for in vivo preclinical toxicological tests to study compounds with potential pharmacological applications in photodynamic therapy. Preclinical testing on this lepidopteran could lead to a significant reduction in the number of experimental animals, as well as offer economic advantages and simpler experimental procedures.

## 4. Materials and Methods

### 4.1. General

All reagents were supplied by Merck KGaA, (Darmstadt, Germany).

5,10,15,20-tetrakis(4-hydroxyphenyl)-porphyrin (also called TPPOH) was dissolved in DMSO to obtain a 10 mM stock solution.

All materials, buffers and solutions were autoclaved or filtered with 0.22 µm Minisart filters (Sartorius, Goettingen, Germany).

*G. mellonella* healthy late-stage larvae (about 300–350 mg) were used for the experimental studies (3 batches of 10 larvae). Larvae were supplied from a rearing facility and kept at 26 °C in a climate chamber in the dark. Before tests, larvae were surface sterilized with 70% ethanol.

A 500 W white tungsten halogen lamp was used for irradiation with an irradiance of 22 mW/cm^2^ equal to 158 J/cm^2^ of fluence for 2 h of irradiation. For this type of lamp, a cooling apparatus is required to avoid overheating; therefore, a running water filter was placed between the light and the irradiation area.

Centrifugations were carried out with a SIGMA 1–14 microcentrifuge (SciQuip Ltd., Newtown, Wem, Shropshire, UK) and Eppendorf 5804 centrifuge (Eppendorf, AG, Hamburg, Germany).

Hemocytes were counted using a Corning Cell counter (Corning Inc., New York, NY, USA) at a magnification of 5X, and the data obtained were processed by CytoSmart^®^ software (Axion Biosystems, Atlanta, GE, USA).

Flow cytometric analyses were conducted using FACScalibur (Becton Dickinson Mountain View, CA, USA) equipped with a 15 mW, 488 nm, air-cooled argon laser and data were analyzed using Cell QuestPRO software (Becton Dickinson).

Cell images were acquired using a camera connected to an Olympus IX51 microscope connected to an OPTIKA mod. C-P20CM digital camera (OPTIKA Srl., Ponteranica, Italy).

Statistical analyses of the data obtained from the various experiments (3 independent tests) were performed by one-way ANOVA. The ANOVA test was performed considering data with a *p*-value less than 0.05 as significant [66].

### 4.2. Phototoxicity Assays

To assess the phototoxicity of TPPOH in *G. mellonella* larvae, 10 μL of TPPOH solution were microinjected into the abdominal spiracle of the larvae with a sterile gas-tight syringe with a 30-gauge hypodermic needle (Hamilton Co, Reno, NE, USA). By diluting stock solution in sterile phosphate buffer saline 1X (PBS), nine concentrations were used (final concentration in larvae range from 1000 to 0.1 µM).

Once injected, larvae were incubated for 24 h in dark conditions, then irradiated for 2 h and once again kept in the dark. Survival rates were recorded at 24, 48 and 72 h after PDT treatment.

To assess possible intrinsic toxicity of TPPOH, larvae were treated with the same concentrations but in this case the irradiation step was skipped.

In addition, as a further control test, larvae were injected with 10 μL of DMSO solution in PBS (100, 500 µM final concentration in larvae) or sterile PBS and then exposed to light or kept in dark, under the same conditions.

Morphologic changes with respect to body darkening and the mortality of larvae were observed. Mortality was defined as complete loss of mobility, verified by physical stimulation with tweezers. LD_50_ values were calculated.

### 4.3. Cytotoxicity Assay and Total Hemocytes Counts

The hemocyte number in *G. mellonella* hemolymph was evaluated. Larvae were injected with the same concentrations used for the toxicity assay, and at fixed time (dark incubation: 24, 48 and 72 h post-injection; PDT treatment: 24, 48 and 72 h after irradiation) hemolymph was collected.

Briefly, larvae were anesthetized on ice, surface sterilized with 70% ethanol and punctured with a sterile needle in the ventral region. The hemolymph was collected in Eppendorf microcentrifuge tubes; to determine cell viability, 10 μL of hemolymph was resuspended with 10 μL of Trypan blue staining solution (1%), then 10 μL of hemolymph were disposed on a Neubauer hemocytometer. Hemocytes were counted using a Corning Cell counter and the data obtained were processed by CytoSmart^®^ software (Axion Biosystems, Atlanta, CA, USA) [67].

For the control, larvae injected with 10 μL of sterile PBS were used.

### 4.4. Cellular Uptake: Flow Cytometric Analysis and Microscopy Technique

To assess the uptake of TPPOH over time (2, 8 and 24 h), *G. mellonella* larvae were injected with 10 μL of 100 μM TPPOH and the intrinsic fluorescence of the PS was exploited to evaluate its accumulation in hemocytes by FACSCalibur flow cytometer using a 575-nm bandpass filter. Briefly, 2, 8 and 24 h post-injection, hemolymph was extracted from *G. mellonella* larvae and hemocytes were obtained by centrifugation at 250× *g* for 5 min at 4 °C. After removal of the supernatant (humoral component), the cell pellet was resuspended in 200 μL of a solution of Grace’ insect medium, containing PTU. Samples were immediately processed by flow cytometer using the CellQuestPRO software (BD Biosciences, San Jose, CA, USA).

Intracellular localization of TPPOH was also assessed by fluorescence microscopy using an in vitro assay. *G. mellonella* larvae were bled at 2, 8 and 24 h post-injection, and hemocytes were isolated. After separation of the cells from the humoral fraction, hemocytes were washed twice with PBS, resuspended in 1 mL of Grace’s insect culture medium plus PTU, plated at a final concentration of 2 × 10^5^ cells/mL, in 24-well plate (Thermofisher Scientific, Waltham, MA, USA) and finally incubated at 26 °C in dark conditions. After 30 min of culture, intracellular storage of TPPOH was observed with an Olympus IX51 microscope connected to a digital camera. Control larvae were injected with 10 μL of sterile PBS.

### 4.5. ROS Detection after PDT Treatment by NBT Reduction

To detect the production of ROS in hemocytes treated with TPPOH and subjected to PDT, a slight modification of the method of Glupov was used [43]. After processing cells as described in Section 4.4, hemocytes were cultured in microwell plates (Corning Inc., New York, NY, USA), with slides, at a concentration of approximately 2 × 10^4^ cells/well in Grace insect medium solution plus 0.1% PTU. To the culture, 1.7 mg/mL NBT and TPPOH (100 µM final concentration) were added. Cells were kept at 26 °C for 24 h in dark conditions. Hemocytes were then subjected to PDT for 2 h or kept in the dark. As a control, cells were cultured with NBT, without TPPOH, and irradiated with light. Hemocytes were washed and fixed for 4 min with PBS buffer containing 1.6% formalin and 0.25% glutaraldehyde, then cells were washed with PBS. Slides were observed with a Zeiss Axiolab microscope (Carl Zeiss, New York, NY, USA) in bright field. Images were recorded with an Optika C-H4K (OPTIKA Srl., Ponteranica, Italy) digital camera.

## Figures and Tables

**Figure 1 ijms-24-03131-f001:**
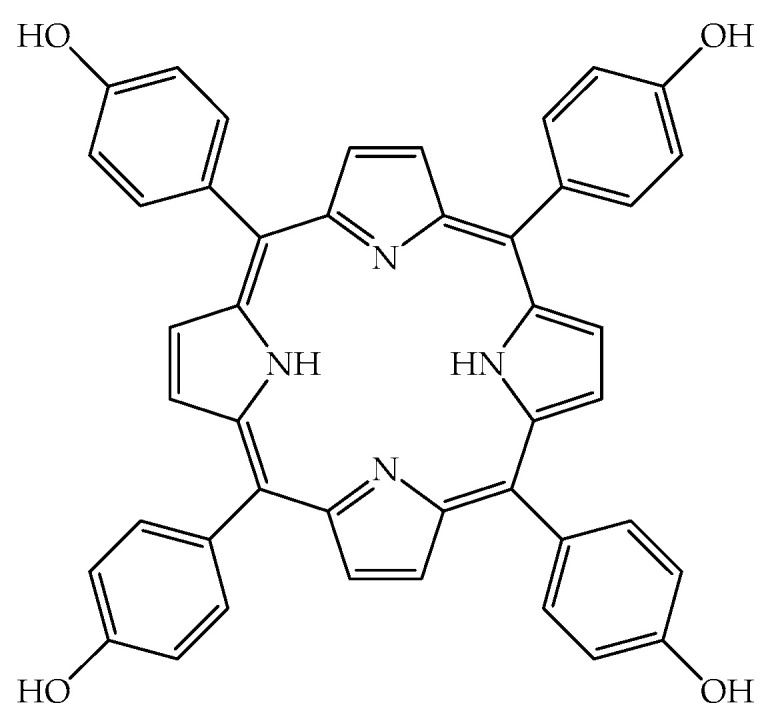
Chemical structure of the 5,10,15,20-tetrakis(4-hydroxyphenyl)-porphyrin (also named as TPPOH).

**Figure 2 ijms-24-03131-f002:**
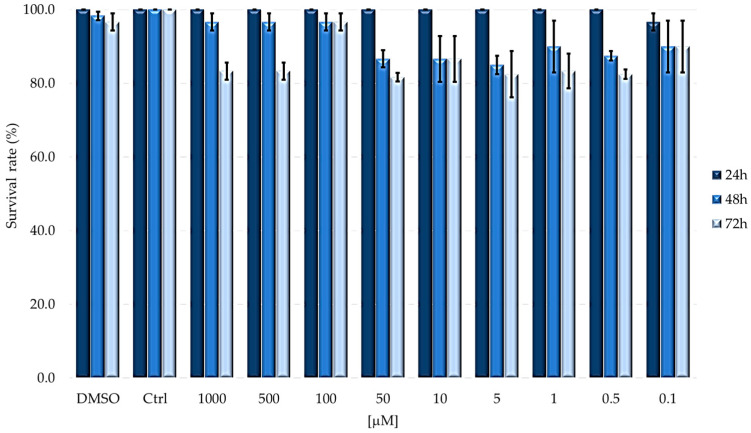
Survival rate of larvae subjected to TPPOH treatments in the dark. Larvae were injected with concentrations ranging from 1000 to 0.1 μM of TPPOH. Survival was evaluated at 24, 48 and 72 h (mean ± SD of 3 independent experiments).

**Figure 3 ijms-24-03131-f003:**
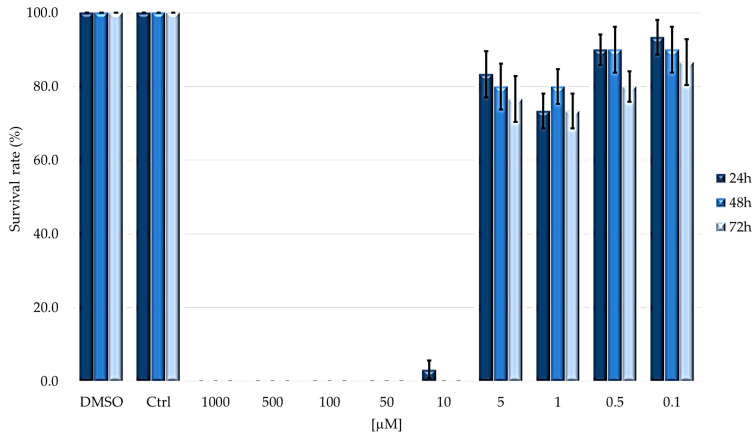
Survival rate of larvae subjected to TPPOH after photodynamic treatment. Larvae were injected with concentrations ranging from 1000 to 0.1 μM of TPPOH. Survival was evaluated at 24, 48 and 72 h after irradiation (mean ± SD of 3 independent experiments).

**Figure 4 ijms-24-03131-f004:**
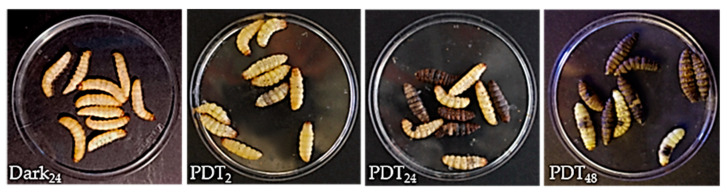
Physiological changes after PDT treatment. Larvae were injected with TPPOH (100 mM) and, after 24 h, irradiated (PDT) for 2 h. Physiological and morphological changes are evident after TPPOH administration and PDT treatment; *G. mellonella* larvae gradually reduce motility, the body becomes floppy and a gradual increase in brownish coloration is observed. After long time periods (24–48 h) post PDT treatment, larvae turn black due to melanization induced by the proPO system activated under stress conditions. Dark_24_: before PDT; PDT_2_: 2 h after PDT; PDT_24_ and PDT_48_: 24 and 48 h after PDT.

**Figure 5 ijms-24-03131-f005:**
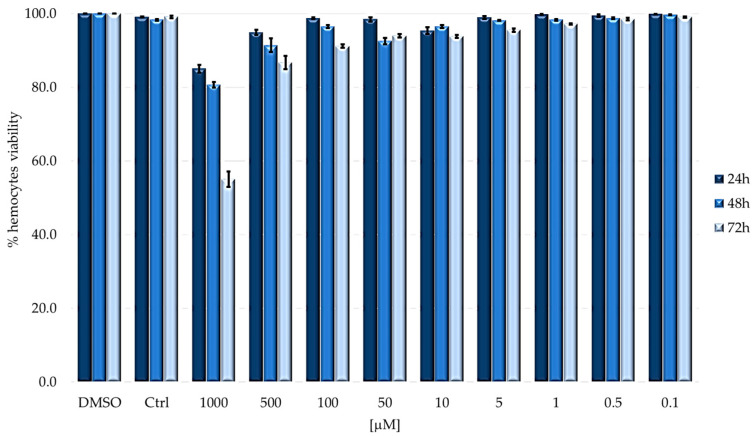
Hemocytes viability of larvae injected with TPPOH under dark conditions. Larvae were injected with concentrations ranging from 1000 to 0.1 μM of TPPOH. Hemocytes viability was evaluated at 24, 48 and 72 h (mean ± SD of 3 independent experiments).

**Figure 6 ijms-24-03131-f006:**
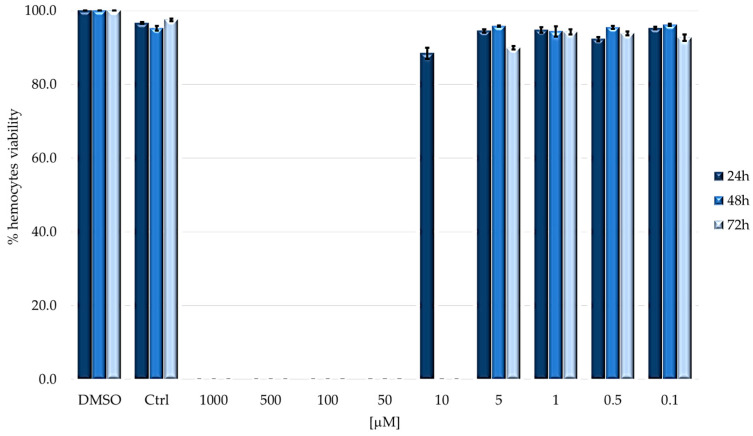
Hemocytes survival rate after photodynamic treatment. Larvae were injected with concentrations ranging from 1000 to 0.1 μM of TPPOH. Hemocytes viability was evaluated at 24, 48 and 72 h (mean ± SD of 3 independent experiments).

**Figure 7 ijms-24-03131-f007:**
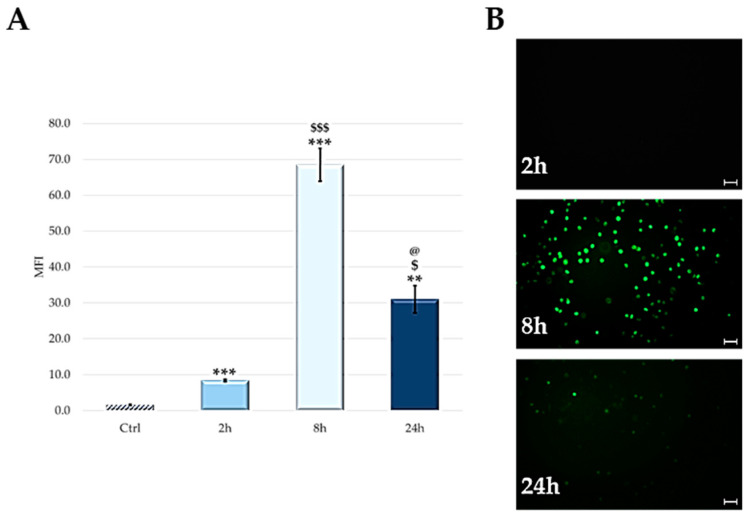
(**A**) Cellular uptake by FACS. Mean ± SD of 3 independent experiments. *** *p* < 0.001, ** *p* < 0.01, vs. Ctrl; $$$ *p* < 0.001, $ *p* < 0.05 vs. 2 h; @ *p* < 0.05 vs. 8 h. (**B**) Cellular uptake evaluated through fluorescent microscopy at 2, 8 and 24 h post-injection (representative images of three Independent experiments). Scale bar: 40 μm.

**Figure 8 ijms-24-03131-f008:**
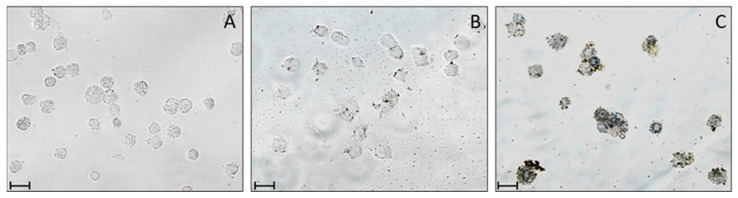
ROS detection in *G. mellonella* hemocytes. *G. mellonella* hemocytes were monitored after TPPOH and NBT addition to cell cultures. Hemocytes have been observed both before and after light irradiation (PDT). Cytoplasmic precipitates are particularly evident in cells after PDT (**C**), a weak reaction is also present in non-irradiated cells (**B**). In contrast, no reaction is observed in irradiated hemocytes treated with NBT without TPPOH. (**A**) control NBT + PDT; (**B**) TPPOH + NBT Dark; (**C**) TPPOH + NBT + PDT. Scale bar: 40 μm.

**Table 1 ijms-24-03131-t001:** LD_50_ values obtained after PDT treatment. Mean + SD of three independent experiments.

	24 h	48 h	72 h
LD_50_ [μM]	5.247 ± 0.932	5.262 ± 1.079	3.936 ± 0.865

## Data Availability

Not applicable.

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
