# Peer review of "Preliminary Toxicity Evaluation of a Porphyrin Photosensitizer in an Alternative Preclinical Model"

_ijms, 2023, doi:10.3390/ijms24043131_

Round 1
Reviewer 1 Report
The manuscript entitled "Preliminary toxicity evaluation of a porphyrin photosensitizer in an alternative preclinical model" described by Malacarne and co-workers describes the evaluation photodynamic activity of tetra-hydroxyphenyl-porphyrin. The subject contemplates the interest of IJMS journal and photochemical readers and can be accepted for publication after major revision in the quality of the manuscript and in the data acquisition:
1. Tests of photostability and stability in porphyrin solution must be conducted, using the same period of irradiation of the experiments;
2. Experiments proving the formation of ROS via spectroscopy or EPR must be performed under the same test conditions for the authors' hypothesis to be valid. They will likely detect the presence of singlet oxygen or superoxide;
3. A time-kill curve follow-up time of 2h should be conducted for better interpretation of light dosage and photodynamic activity;
4. Authors should explain in the introduction why they chose TPOHP. The reason for its use is unclear;
Author Response
Dear Reviewer,
We thank you for the valuable and stimulating observations. Your suggestions and criticisms further contributed to improve the quality of the paper. We answer point by point to your main concerns.
1) Tests of photostability and stability in porphyrin solution must be conducted, using the same period of irradiation of the experiments.
4) Authors should explain in the introduction why they chose TPOHP. The reason for its use is unclear.
As requested by the reviewer, the authors explain the use of TPPOH in the introduction part (lines 88-91 instead of lines 41-43 that were removed). In this part authors added two references (ref 41-42) for the characterization and photostability analysis of the TPPOH. Due to the changes, Figure 1 was moved at the end of the introduction.
2) Experiments proving the formation of ROS via spectroscopy or EPR must be performed under the same test conditions for the authors' hypothesis to be valid. They will likely detect the presence of singlet oxygen or superoxide.
To determine the ROS formation in 2D haemocytes cell cultures the authors performed a new experiment which is a modification of a procedure reported by Glupov (2001) (see paragraph 4.5). The authors used NBT to detect ROS formation and bright-field microscopy to visualize NBT-derived formazan salts in cells. Results were reported in lines 178-200 and Figure 8. In lines 276-284 authors discussed them.
3) A time-kill curve follow-up time of 2h should be conducted for better interpretation of light dosage and photodynamic activity.
As can be seen in the new added figure, relating to the morphological state of the larvae after 2, 24 and 48h from the end of the illumination (Figure 4), during the first 2h no morphological changes were observed and all the larvae were alive and had good mobility. This suggests that the light dosage and the 2h irradiation do not cause damage to the larvae.
Sincerely yours
Enrico Caruso
Reviewer 2 Report
This is a very interesting and well written manuscript. It introduces a new and promising in vivo method for testing the photodynamic effectiveness and dark toxicity of new photosensitizers without the use of expensive, time consuming and ethically questionable mammalian models. My only suggestion would be to also discuss the appearance of the live and dead larvae with some photos added; i.e. does PDT-induced death cause any morphological changes in the larvae? Perhaps a dose-response change in larvae appearance is observable.
Author Response
Dear Reviewer,
We thank you for the valuable and stimulating observations. Your suggestions and criticisms further contributed to improve the quality of the paper.
My only suggestion would be to also discuss the appearance of the live and death larvae with some photos added; i.e. thus PDT induce death cause any morphological changes in the larvae? Perhaps a dose-response change in larvae appearance is observable.
The authors added Figure 4 in which was reported the larvae morphological state after 2, 24 and 48h from the end of the irradiation. In lines 131-135 authors explained that no morphological changes appeared in the first 2h after irradiation while after 24 or more hours variations in larvae morphology were observed with a decrease in turgidity and progressive darkening (blackish color).
Sincerely yours
Enrico Caruso
Round 2
Reviewer 1 Report
The revised manuscript can be accepted in its present form.